# Fabrication of Bio-Based Film Comprising Metal Oxide Nanoparticles Loaded Chitosan for Wound Dressing Applications

**DOI:** 10.3390/polym15010211

**Published:** 2022-12-31

**Authors:** Latifah Mohammed Ali Almaieli, Mai M. Khalaf, Mohamed Gouda, Abraham Elmushyakhi, Manal F. Abou Taleb, Hany M. Abd El-Lateef

**Affiliations:** 1Department of Chemistry, College of Science, King Faisal University, Al-Ahsa 31982, Saudi Arabia; 2Chemistry Department, Faculty of Science, Sohag University, Sohag 82524, Egypt; 3Department of Mechanical Engineering, College of Engineering, Northern Border University, Arar 91431, Saudi Arabia; 4Department of Chemistry, College of Science and Humanities, Prince Sattam Bin Abdulaziz University, Al-kharj 11942, Saudi Arabia; 5Department of Polymer Chemistry, National Center for Radiation Research and Technology (NCRRT), Egyptian Atomic Energy Authority, Nasr City, P.O. Box 7551, Cairo 11762, Egypt

**Keywords:** wound dressing, chitosan, polymer nanocomposites, metal oxide nanohybrid, biomedical applications

## Abstract

In the current work, chitosan (CS)–metal oxide nanohybrid (MONH) composites are prepared via combining CS with MONH made of vanadium oxide (V_2_O_5_), ytterbium trioxide (Yb_2_O_3_), and graphene oxide (GO) to generate promising wound dressing materials using the film-casting method. The developed nanohybrid@CS was examined using techniques such as Fourier-transform infrared spectroscopy (FTIR), X-ray powder diffraction (XRD), scanning electron microscopy (SEM)/energy-dispersive X-ray spectroscopy (EDX), and thermogravimetric analysis (TGA). For Yb_2_O_3_@CS, the surface morphology was shown to be a rough and porous surface with pores that ranged in size from 3.0 to 5.0 µm. For CS with Yb_2_O_3_, Yb_2_O_3_/V_2_O_5_@CS, and Yb_2_O_3_/V_2_O_5_/GO@CS, the contact angles were 72.5°, 68.2°, and 46.5°, respectively. When the nanohybrid@CS was in its hydrophilic phase, which is good for absorbing moisture and drugs, there was a notable decrease in angles that tended to rise. Additionally, the inclusion of MONH allowed the cell viability to be confirmed with an IC_50_ of 1997.2 g/mL and the cell growth to reach 111.3% at a concentration of 7.9 g/mL.

## 1. Introduction

Wounds have become a major problem all over the world. For example, the United States recorded an increase in wounds of about 71% during the period from 2012 to 2018 [1]. A wound occurs in the skin layer of the human body, which is a barrier between the internal organs and external factors [2]. Injuries, burns, surgical damage, sharp objects, and accidents are the most common reasons for harmful wounds [3,4]. Wound healing is an essential process for restoring damaged tissues in a particular period. The overlapping phases, hemostasis, inflammation, proliferation, and tissue remodeling are the main stages of the healing process [5]. One of the most important requirements to accelerate the healing process is to develop wound bandages by using polymeric biomedical materials that have biocompatible behavior and suitable flexibility [6].

Chitosan (CS) is a natural biopolymer with good biocompatibility and biodegradable behavior; however, it has poor mechanical properties [7]. It represents multi-benefit properties such as anti-inflammatory, biological adhesion, and hemostatic properties [8]. It can be applied in biomedical applications such as drug delivery, tissue engineering, and wound healing dressings [9].

Ytterbium trioxide or ytterbia (Yb_2_O_3_) is a rare-earth oxide with high thermal stability, which leads to its common usage in material science studies [10]. Vanadium oxide (V_2_O_5_) is considered a transition metal oxide [11]. It is suitable for several applications, electrical and biological, due to its low toxicity, antimicrobial properties, and biocompatible behavior [12]. Graphene oxide (GO) has common usage in the nanotechnology field and other studies due to its two-dimensional honeycomb structure [13,14]. It exhibits several suitable properties such as high mechanical strength, low preparation cost, and good hydrophilicity [13,15,16]. The aim of this study is to enhance wound dressings with additional properties by using CS with different contributions of Yb_2_O_3_ and V_2_O_5_. In addition, the excellent mechanical strength of metal oxides may recover the decrease in mechanical properties in CS. A study reported by Y. Wanga et al. showed CS fabricated with poly(lactic-co-glycolic acid) (PLGA) and tylotoin nanoparticles to improve the drug delivery system and promote the wound healing process. The results show that CPT nanoparticles can enhance the wound healing cycle, inhibit bacterial growth, and accelerate the proliferative phases [8].

A study described by J. Movaffagh et al. presented the development of CS with aloe vera hydrogel, and the results confirmed the excellent reduction in inflammation and suitable acceleration in wound healing and body recovery [17]. In this study, novel wound dressings based on CS film containing hybrid nanocomposites, including Yb_2_O_3_, V_2_O_5_, and GO, are targeted. CS is thought to play a key role as a good matrix biomaterial for drug encapsulation. On the other hand, V_2_O_5_ is drugged in the CS film to fit with the targeted application as it has low toxicity, high antibacterial activity, and high biocompatibility, while Yb_2_O_3_ and GO are used to support the biological response of the scaffolds and to improve the antibacterial properties. These combinations of nanocomposites are drugged in the CS film, aiming to improve the physical and chemical properties, bioactivity, and cell viability of the designed scaffolds so that they are appropriate for medical applications. The main structural and morphological features of the obtained films are studied, while the surface wettability and the cell viability compared to normal cells are investigated. The antibacterial properties of the designed scaffolds of CS containing different contributions of Yb_2_O_3_, V_2_O_5_, and GO are evaluated against Gram-positive and Gram-negative pathogens to select the optimum compositions for medical applications.

## 2. Materials and Methods

### 2.1. Materials

The basic materials used were ytterbium trioxide and vanadium oxide, graphite, hydrochloric acid (HCl), potassium permanganate (KMnO_4_), sulphuric acid (H_2_SO_4_), H_3_PO_4_, H_2_O_2_, chitosan (M = 40,000 g/mol), and acidic acid. These materials were obtained from Sigma-Aldrich Co., St. Louis, MO, USA.

### 2.2. Fabrication CS Thin Film Containing Metal Oxide Hybrid Nanocomposites

Yb_2_O_3_ and V_2_O_5_ nanoparticles were used as they were obtained. Graphene oxide (GO) was fabricated in the lab using the modified Hummer method. A 3 g amount of graphite powder was added to a 9:1 mixture of H_2_SO_4_/H_3_PO_4_ concentrate during continuous stirring for 5 min and then 18 g of KMnO_4_ was added to the previous blend and stirred continuously for 12 h. Moreover, 3 mL of H_2_O_2_ was added and mixed for 1 h. The solution was washed with 30% HCl, distilled water, and ethanol and then dried in the furnace. CS thin film containing metal oxide hybrid nanocomposites was prepared by dissolving 7 wt.% CS, and 5 samples of CS were prepared in stock, each containing 20 mL. The first was pure CS with no additives. Metal oxide hybrid nanocomposites were added to another CS solution at a total weight of 0.25 g per film. It should be noted that the addition was dropped into each bottle with stirring using a magnetic stirrer for 1 h to obtain well-dispersed solutions. The samples were molded into a Petri dish and dried in an oven.

### 2.3. Characterizations

X-ray diffraction (XRD) was performed (with an analytical Pertpro with Cuk_α1_, The Netherlands). The scan was carried out from 10 degrees to 70 degrees. The step size was about 0.5 s. The wavelength of the X-ray beam was 1.54 Å. Fourier-transform infrared (FTIR) spectra were used for all samples (Perkin-Elmer (FTIR) in the range of 4000–400 cm^−1^). The samples were prepared for the test by pressing the powdered/crushed film with KBr with a ratio of 20:1. Then, the samples were fixed on the holder ready for the test. The morphology of the surface was investigated using a scanning electron microscope (SEM) (QUANTA-FEG250, The Netherlands). The operating voltage was around 8 kV. The thermal decomposition temperature of chitosan nanocomposites was determined by thermal gravimetric analysis (TGA) (STD Q600) from 25 to 800 °C at a heating rate of 10 °C/min in nitrogen medium.

#### Examination of Films Morphology

FESEM was used to investigate the morphology of normal cells on the films-based nanocomposites. For this issue, the nano-film was sanitized using a UV lamp with 30 min of exposure. Each sample was cropped into two pieces of 0.5 × 0.5 cm, then they were inserted into 12-well plates. A 1.5 mL amount of normal cells was added to each well. The plate was then incubated at 37 °C for 3 days. After this time, the films were washed with phosphate-buffered saline (PBS). To keep the cells fixed on the film surface, the scaffolds were submerged in a glutaraldehyde solution (4% concentration) for 1.0 h. Then, they were dehydrated in the air for ¼ h. Finally, they were coated with gold for just 2 min to be ready for FESEM resultant surface images.

### 2.4. Contact Angle

The contact angle was evaluated using a customized system of water drops. A sample of 1 cm^2^ of each sample was fixed versus the camera, and the images were taken when the drop of water was released.

### 2.5. In Vitro Cell Viability Tests

The cell viability was tested with normal lung cells. The culture was in Gibco medium. The samples were weighed separately and then soaked in sterilized water for 24 h. The solution was serialized in 96-well plates from the highest concentration to the lowest level of the scaffold. Then, the plates were incubated for 72 h at 37 °C. Following this, media were removed, and MTT was added to measure cell viability using optical density. The cell lines’ data were as follows: accession numbers: WI-38 (ATCC CCL-75); database name: ATCC. The WI-38 cell lines were isolated from the lung tissue of a three-month-old female embryo. Organism: Homo sapiens, human; cell type: fibroblast; tissue: lung; age: 3 months’ gestation; gender: female; morphology: fibroblast; growth properties: adherent; disease: normal.

### 2.6. Antibacterial Test

The antibacterial behavior was carried out against two species: *Escherichia coli* (*E. coli*) and *Staphylococcus aureus* (*S. aureus*). The initial concentration of the samples was fixed at 50 mg/mL. In detail, a fixed weight (100 mg) of each sample was added into 2 mL of deionized water to be immersed for 1 day before the test. Then, each sample was put into its position in the Petri dish to be exposed to bacterial cells for 24 h.

## 3. Results and Discussion

### 3.1. XRD

Figure 1 shows the X-ray diffraction graph for CS with Yb_2_O_3_ and V_2_O_5_. It was used to analyze the crystal structure and the degree of crystallinity. Pure CS had a broad peak at 2θ = 19.8°, which refers to the semicrystalline phase of CS [18]. The diffraction peaks of Yb_2_O_3_ were determined at 2θ = 20.5, 29.3, 34.1, 44.06, 49.1, 53.9, and 58.4°, which correspond to Miller’s indices (211), (222), (400), (622), (440), (611), and (622) [19]. The high, intense peak was at 29.3; however, peaks at 49.1 and 58.4° were ascribed as weak diffraction peaks that refer to the cubic structure of Yb_2_O_3_ [20]. In addition, V_2_O_5_ exhibited several peaks at 2*θ* = 15.08, 20.6, 26.03, and 30.7°, which correspond to the orthorhombic structure of V_2_O_5_ with Miller’s indices (200), (001), (110), and (301) [21].

### 3.2. FTIR

The FTIR spectra of CS films with nanoparticles Yb_2_O_3_ and V_2_O_5_ are shown in Figure 2 in the range of 500–4000 cm^−1^. They were used to determine the entire structure of each film based on the vibration bonding systems. For pure CS, the bands at 658, 894, and 994 cm^−1^ might refer to N–H bonds, the glucosidic C–O–C stretching vibration, and C–O–C symmetrical stretching, as shown in Table 1 [22,23,24]. The recorded bands at 1091, 1152, and 1334 cm^−1^ may represent the C–O stretching vibration and –OH group of the primary alcoholic group [12,22,25]. In addition, the determined bands at 1407, 1538, and 1555 cm^−1^ exhibit the –OH group of the primary alcoholic group, the symmetric bending of -NH^+3^, and amide II (N–H deformation present in the amino group) [22,23,26]. Bands at 2878 and 3397 cm^−1^ might be assigned to the CH stretching vibration and the overlaps between primary amine -NH and the OH group [23,26]. In Yb_2_O_3_@CS, the bands at 472, 611, 1512, and 1649 cm^−1^ were ascribed to Yb–O bonds, stretching vibrational modes of the C–O bond, and the C–O stretching vibration, respectively [27,28]. In the range of 3000–4000 cm^−1^, the O–H stretching vibrations were false [27]. The bands at 472, 611, and 613 cm^−1^ seem to represent the vibrational bond of Yb–O [27]. On the other hand, the vanadium oxide bonding system was found in a band of 558 cm^−1^ [29]. It should be mentioned that the bonds of 658, 1334, and 1538 cm^−1^ might prove the contribution of CS through the composition. In addition, the bands of O–H seem to have been lowered from the pure CS to the modified ones, which were assigned to the prospective interactions between the CS and the doped particles, especially the GO nanosheets. The interaction might have occurred via initiating hydrogen bonding mechanisms.

### 3.3. Surface Morphology

The morphology of the CS and the nanocomposites was analyzed with a scanning electron microscope (SEM). Figure 3a,b shows the change in the surface topography of CS with Yb_2_O_3_. The micrograph appears to be rough with several micropores and noticeable micro-cracks which have diameters varying between 3 µm and 5 µm. It was confirmed that pores have an essential role in moisture, gases, and drug transporting through wounds and the outer environment. Moreover, the porous surface might improve the mutual equilibrium between the mechanical and biological properties [30].

The scattered shapes on the surface of CS indicate the good distribution of Yb_2_O_3_ according to its high crystallinity_._ On the other hand, the tendency for roughness might enhance the adhesive state of the scaffolds. As shown in Figure 3c,d, the surface tended to be smooth, which may decrease the adhesion behavior and low porosity and which may limit the moisture and gas exchange due to the addition of V_2_O_5_. In Figure 3e,f, it can be seen that the porosity started to appear slightly with a diameter in the range of 0.5–2 µm with low roughness due to the addition of GO nanosheets that had low crystallinity and a 2D structure.

In the SEM micrographs by S. Aggarwal et al., the surface topography was represented as the porous surface of ZnO/chitosan with major roughness due to the rough matrix of CS, which might lead to a large surface area [23].

### 3.4. EDX

The term EDX refers to the energy-dispersive X-ray test, which was used to detect the different ratios of the contributed elements (Figure 4). As shown in Figure 4 (table inset), the determined high ratios referred to carbon and oxygen at 61.41 and 36.47% and were ascribed to the major concentration of CS. In addition, the EDX profile of the nanocomposites showed identified ratios for V and Yb of about 0.73 and 1.39%, respectively. It should be noted that the nanocomposites were detected on the surface of CS by using SEM.

### 3.5. Contact Angle

Contact angle characterization measures the hydrophilicity of scaffolds due to the degree of wettability. The surface of a scaffold with a contact angle greater than 90° is generally considered to be hydrophobic; otherwise, it is hydrophilic. As is obvious from Figure 5, the average contact angle of Yb_2_O_3_@CS was 72.5°. This angle exhibits low hydrophilic characteristics, which might limit cell adhesion and proliferation. With the addition of V_2_O_5_, the average angle tended to slightly decrease to 68.2°, which resulted in an increase in the hydrophilicity of the surface. After adding GO, the angle experienced a significant decrease, reaching 46.5°, which might be due to the hydrophilic outer surface of GO, which can enhance cell attachment, cell growth, and the adhesive behavior of the scaffolds; thus, it is beneficial to develop the hydrophilicity. In addition, it is clear that by adding the nanocomposites and GO to the CS films, the hydrophilicity of the surface was increased when the contact angle was reduced. The insertion of GO nanosheets through the composition might have promoted the formation of hydrogen bonding between the connected oxyanions over the graphene sheets with the graphene oxides, which was mentioned in the discussion of the FTIR study. The formation of chemical bonds between the two constituents might have developed rough surfaces, as shown in the SEM graphs. These topographical changes might be essential for promoting the hydrophilic affinity of the obtained composites.

### 3.6. Thermal Stability

The thermal stability of materials was determined by the thermogravimetric analysis (TGA) test. Figure 6 shows a degradation of Yb_2_O_3_/ V_2_O_5_/GO@CS scaffolds through different stages. The first stage is represented by 24.3–256.4 °C, with a noticeable decrease in the weight due to the water or acetic acid vaporization process on the surface of the CS films. The second stage exhibits mass loss of about 52.3% in the range of 256.4–462.9 °C, which might refer to the molecular decomposition of the film or the breaking of the H bonds [31].

The last degradation stage occurs at 462.9–597.3 °C, with a total weight loss of about 79%, which indicates the complete decomposition of the CS film and the other additives of nanocomposites. A study by P. Amin et al. showed that CS has initial mass loss at the stage of 30–150 °C due to the water evaporating, while the significant weight loss occurs at 300 °C due to the complete thermal degradation of the CS unit [32].

### 3.7. Cell Viability

Cell viability is one of the most important factors used to determine the compatibility of implant material with normal cells for 3 days of culturing. As is obvious from Figure 7, the scaffold of Yb_2_O_3_/V_2_O_5_/GO@CS was started at an initial concentration of about 19,986.3 µg/mL. The IC_50_ is responsible for detecting the degree of toxicity of the loaded drugs at 50%, and it was measured to be 1997.2 µg/mL. Moreover, the addition of metal oxides led to cell growth, reaching 111.3% at a concentration of 7.9 µg/mL. In addition, in Figure 8, the viable cells that were detected by the optical microscope were represented in the form of rod shapes with high distribution, while the dead cells were distributed with low concentration in the form of spherical shapes. In addition, the porous scaffold determined by SEM caused an increase in the cell viability value [30].

### 3.8. Antibacterial Behavior

The ability of the scaffolds to inhibit the prospective invasion of bacteria is one of the most important properties that should be available to avoid delaying the healing procedure. In this regard, antibacterial behavior against Gram-positive and Gram-negative bacterial cells was carried out for all under-investigated compositions to evaluate their strength against the two types of bacteria. As shown in Figure 9, the potency of degenerating bacterial cells increased from that of the pure CS with the additional nanoparticles. The inhibition zone increased from 11.5 ± 0.5 to 19.5 ± 0.5 mm against *E. coli* for pure CS and Yb_2_O_3_ /V_2_O_5_@CS, respectively. On the other hand, the strength of the nanocomposites against *S. aureus* seemed to be slightly higher than against *E. coli*. The mortality of *S. aureus* increased from an inhibition zone of 12.5 ± 0.5 to 19.0 ± 1.0 mm for CS and Yb_2_O_3_/V_2_O_5_@CS. The additional GO tended to plunge the antibacterial efficiency, whereas the inhibition zones decreased to 14.5 ± 0.5 and 13.5 ± 0.5 mm against *E. coli* and *S. aureus*, respectively. The integration of Yb_2_O_3_ and V_2_O_5_ nanoparticles through the scaffolds might be the main parameter inducing the antibacterial potency due to the launching of reactive oxygen species with the toxic fragments against the bacterial cells. Nevertheless, pure GO has its own antibacterial properties; its addition was not reflected by an improvement in the antibacterial efficacy. This unexpected behavior might be associated with the high potential of other nanoparticles to degenerate the bacterial cells compared with the GO. In other words, the additional GO on the account of Yb_2_O_3_ and V_2_O_5_ was not able to compensate for their potency against the bacterial cells with the same concentration.

## 4. Conclusions

As a result of this study, chitosan (CS)–metal oxide (Yb_2_O_3_ and V_2_O_5_) nanohybrid (MONH) composite films were fabricated as wound bandages using the film-casting method in vitro. The micrographs of CS with Yb_2_O_3_ by SEM presented a rough surface with different pores with a diameter between 3 µm and 5 µm, while the addition of V_2_O_5_ reduced the roughness, and the scaffold became smooth, and the pores started to appear again after the addition of GO with diameter in the range of 0.5–2 µm. In addition, the thermal stability of the scaffolds was approved by the TGA curve where the complete degradation of the sample occurred at 462.9–597.3 °C with a total mass loss of about 79%. Moreover, the contact angle of CS with Yb_2_O_3_ was measured at 72.5°, while the addition of V_2_O_5_ led to a slight increase in the hydrophilic behavior, and the angle reached 68.2°, and the significant increase of hydrophilicity occurred when the GO nanosheet was added, and the angle was determined at 46.5°. The cell viability was examined versus normal cells, and it was found that the IC_50_ was 1997.2 µg/mL. In addition, the metal oxides incorporated through the scaffolds led to cell growth, reaching 111.3% at a concentration of 7.9 µg/mL. An antibacterial test was carried out against *E. coli* and *S. aureus*. It was shown that the inhibition zone increased from 11.5 ± 0.5 to 19.5 ± 0.5 mm against *E. coli* for pure CS and Yb_2_O_3_/V_2_O_5_@CS, respectively. Furthermore, the inhibition zone of *S. aureus* increased from 12.5 ± 0.5 to 19.0 ± 1.0 mm for CS and Yb_2_O_3_/V_2_O_5_@CS.

## Figures and Tables

**Figure 1 polymers-15-00211-f001:**
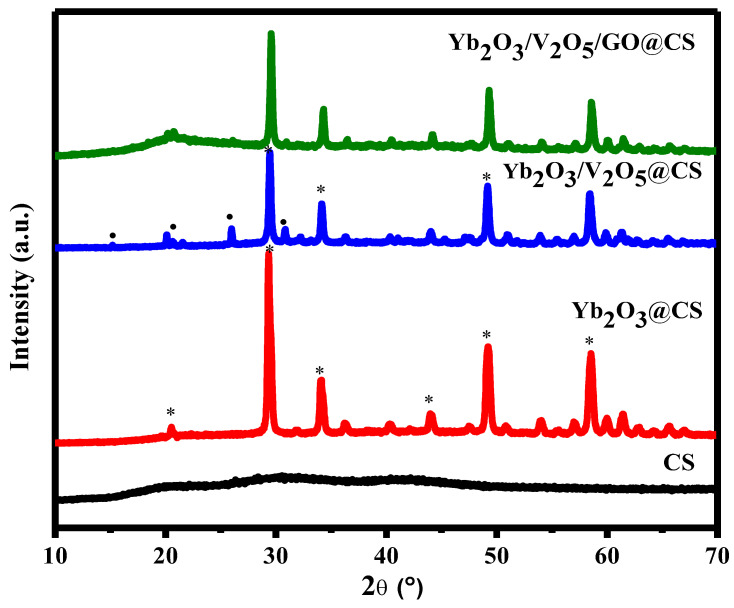
XRD of CS with different ratios of nanocomposites with Yb_2_O_3_ and V_2_O_5_ (* represents the peaks of V_2_O_5_, while the ● indicates the peaks of Yb_2_O_3_).

**Figure 2 polymers-15-00211-f002:**
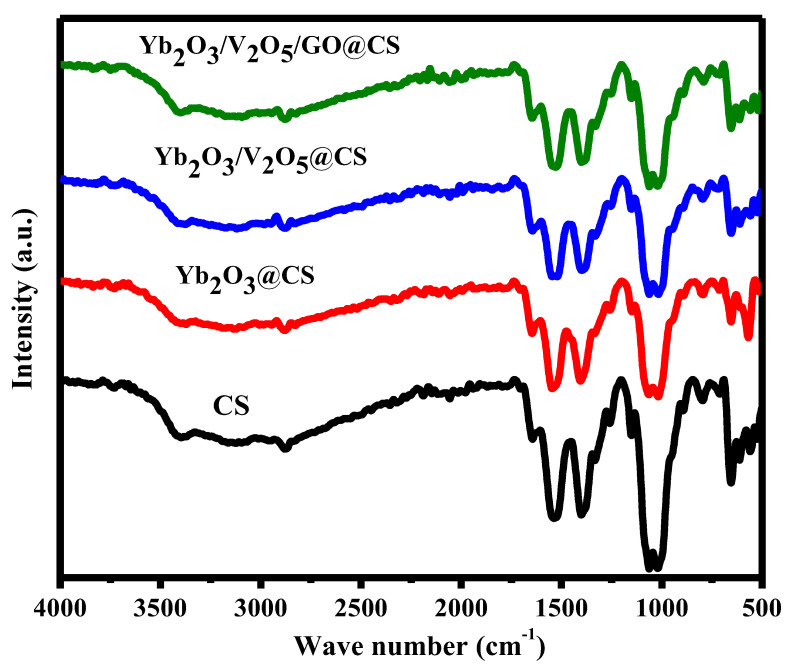
FTIR of CS films with a different contribution of nanoparticles.

**Figure 3 polymers-15-00211-f003:**
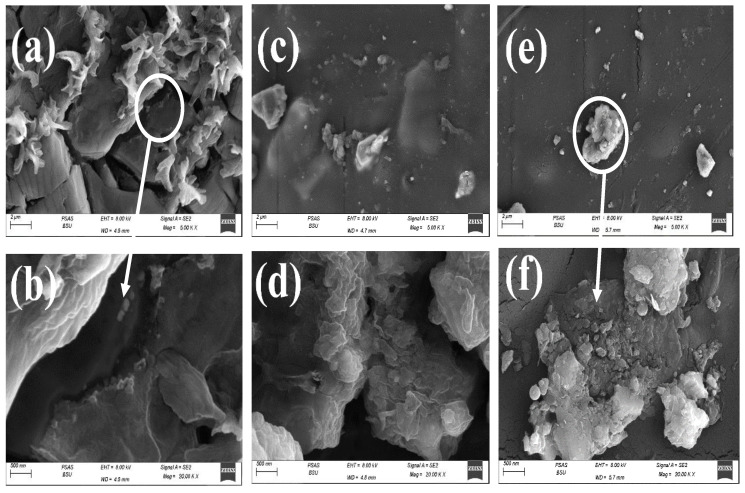
SEM images of: (**a**,**b**) Yb_2_O_3_@CS, (**c**,**d**) Yb_2_O_3_/V_2_O_5_@CS, (**e**,**f**) Yb_2_O_3_/V_2_O_5_/GO@, CS.

**Figure 4 polymers-15-00211-f004:**
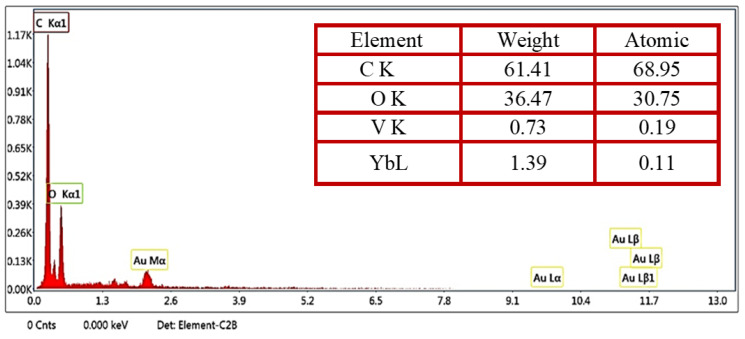
EDX of CS films that contain a different concentrations of Yb_2_O_3_/V_2_O_5_/GO@CS.

**Figure 5 polymers-15-00211-f005:**
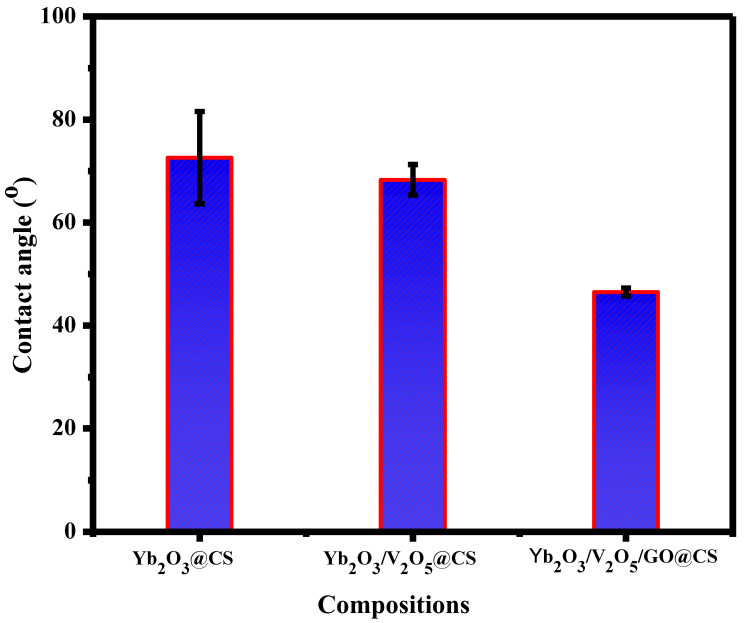
Contact angles measurements of Yb_2_O_3_@CS, Yb_2_O_3_/V_2_O_5_@CS, and Yb_2_O_3_/V_2_O_5_/GO@CS.

**Figure 6 polymers-15-00211-f006:**
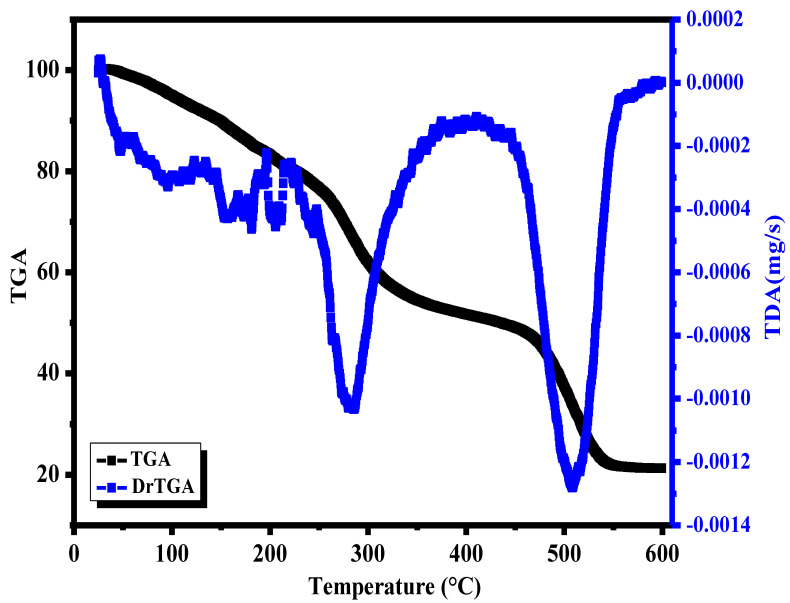
TDA and TGA graph for Yb_2_O_3_/V_2_O_5_/GO@CS.

**Figure 7 polymers-15-00211-f007:**
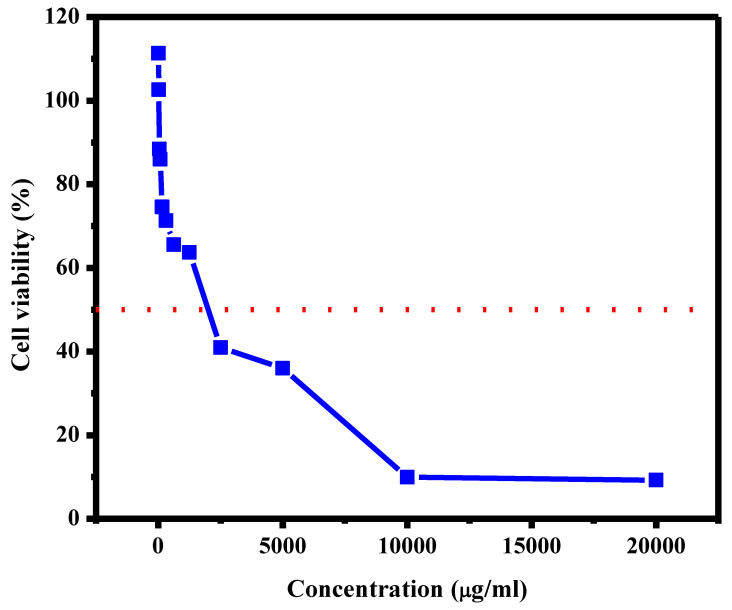
Determination of IC_50_ of viable cells of CS cast film with metal oxides.

**Figure 8 polymers-15-00211-f008:**
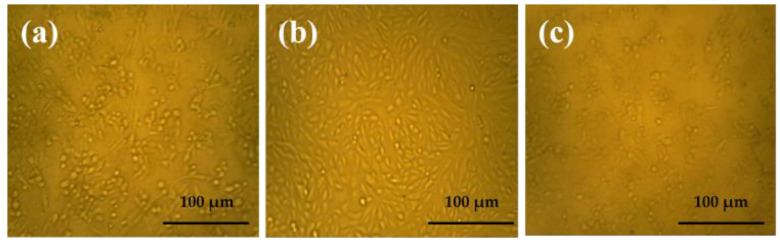
The microscopic images of viable cells of CS films at Yb_2_O_3_/V_2_O_5_/GO@CS, (**a**) the cells with a film concentration of 20 mg/mL, (**b**) at a concentration of 5 mg/mL, (**c**) at a concentration of 2 m/mL (scale bar = 100 µm).

**Figure 9 polymers-15-00211-f009:**
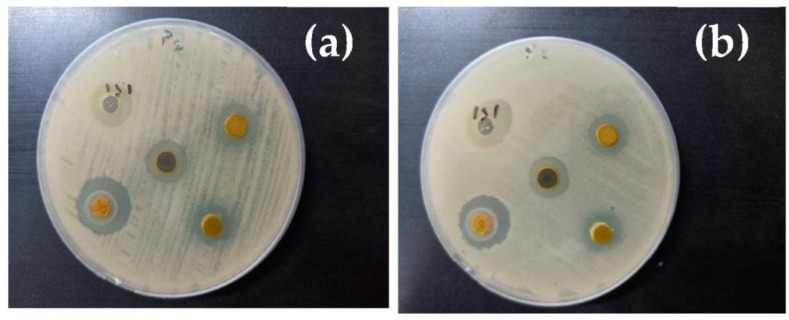
Antibacterial effect of the scaffold films of CS doped with different nanocomposites such as Yb_2_O_3_, V_2_O_5_, and/or GO against (**a**); *E. coli* and (**b**); *S. aureus* after 24 h of exposure.

**Table 1 polymers-15-00211-t001:** FTIR spectra of CS films with nanocomposites in different concentrations.

CS	Yb_2_O_3_@CS	Yb_2_O_3_/V_2_O_5_@CS	Yb_2_O_3_/V_2_O_5_/ GO@CS	Assignment	Ref.
…	472	…	…	Yb–O	[27]
…	…	558	…	V–O–V bond	[29]
…	…	611	613	Yb–O	[27]
658	654	657	656	N–H	[22]
894	…	…	893	C–O–C	[23]
994	…	…	…	C–O–C	[24]
1091	…	…	…	C–O	[25]
1152	1150	1152	1152	C–O	[12]
1334	…	1333	1333	C–O–N group	[22]
1407	1410	…	1407	–OH group	[22]
…	1512	1513	1513	C–O bond	[28]
1538	…	…	…	Bending of -NH^+3^	[26]
1555	1554	1555	1554	Amide II (N–H deformation)	[23]
…	1649	1650	1651	C–O	[28]
2878	2882	2879	2878	CH	[23]
…	…	3120	…	O–H	[27]
…	3131	…	…	O–H	[27]
…	…	3175	3176	O–H	[27]
…	…	3375	…	O–H	[27]
3397	…	…	3399	Amine -NH and OH	[26]

## Data Availability

The raw/processed data generated in this work are available upon request from the corresponding author.

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
