# Peer review of "Fabrication of Bio-Based Film Comprising Metal Oxide Nanoparticles Loaded Chitosan for Wound Dressing Applications"

_polymers, 2022, doi:10.3390/polym15010211_

Round 1

Reviewer 1 Report

In this manuscript, the authors combine chitosan (CS) with metal oxide nanohybrid (MONH) made of vanadium oxide (V2O5), ytterbium trioxide (Yb2O3), and graphene oxide (GO) to prepare chitosan CS-MONH composites using film casting method. FTIR, XRD, SEM/EDX and TGA are used to examine the properties of nanohybrid@CS. The inclusion of MONH shows an excellent anti-bacterial performance, suggesting the hybrid films to be a promising wound dressing material. The results are interesting and give an example of novel wound dressing nanocomposites. The paper is recommended for publication in Polymers. Some comments are suggested below.

(1)    In page 6, SEM images of Yb2O3@CS, Yb2O3/V2O5@CS and Yb2O3/V2O5/GO@CS are showed in Figure 3. Why are EDX images not performed on the same areas? So that the distribution of element contents on the surface could directly be identified.

(2)    In line 69-71, the authors describe “In this study, a novel wound dressings based on CS film containing hybrid nanocomposites to improve the mechanical properties, bioactivity, and cell viability.” As a wound dressing material, the prepared nanocomposites should have a good anti-bacterial property. Some related works, such as Adv. Mater. 34, 2202180 (2022), are suggested. In addition, drug treatments for wounds should also be considered, such as J. Biomed. Nanotechnol. 18, 660 (2022).

(3)    The authors mainly characterize the prepared nanocomposites and only show the anti-bacterial performance. Other characterizations, such as wound hemostasis and wound healing reported in Nat. Commun. 12, 3613 (2021), are suggested.

(4)     What are the differences between (a), (b) and (c) in Figure 8. Are they taken at different areas on a same sample or on different samples? Figure 8 lacks scale bars.

(5)     The manuscript should further be improved. Is the unit of the x axis in Figure 7 “mg/ml” or “μg/ml”?

Author Response

Reviewer 1

In this manuscript, the authors combine chitosan (CS) with metal oxide nanohybrid (MONH) made of vanadium oxide (V2O5), ytterbium trioxide (Yb2O3), and graphene oxide (GO) to prepare chitosan CS-MONH composites using film casting method. FTIR, XRD, SEM/EDX and TGA are used to examine the properties of nanohybrid@CS. The inclusion of MONH shows an excellent anti-bacterial performance, suggesting the hybrid films to be a promising wound dressing material. The results are interesting and give an example of novel wound dressing nanocomposites. The paper is recommended for publication in Polymers. Some comments are suggested below.

(1)    In page 6, SEM images of Yb2O3@CS, Yb2O3/V2O5@CS and Yb2O3/V2O5/GO@CS are showed in Figure 3. Why are EDX images not performed on the same areas? So that the distribution of element contents on the surface could directly be identified.

Response: In the EDX test usually different positions are taken into consideration rather than one site. In detail, the elemental distribution on the sample surface does not be highly homogenous, therefore, different sites can be imaged using EDX, then the average of the obtained values can be calculated to be indexed.

(2)    In line 69-71, the authors describe “In this study, a novel wound dressings based on CS film containing hybrid nanocomposites to improve the mechanical properties, bioactivity, and cell viability.” As a wound dressing material, the prepared nanocomposites should have a good anti-bacterial property. Some related works, such as Adv. Mater. 34, 2202180 (2022), are suggested. In addition, drug treatments for wounds should also be considered, such as J. Biomed. Nanotechnol. 18, 660 (2022).

Response: Thank you for your comment. The mentioned references have been now cited in the manuscript.

(3)    The authors mainly characterize the prepared nanocomposites and only show the anti-bacterial performance. Other characterizations, such as wound hemostasis and wound healing reported in Nat. Commun. 12, 3613 (2021), are suggested.

Response: The mentioned reference has been provided in the manuscript in an appropriate position. However, this study aims to evaluate the morphological changes, and the thermal stability besides the wettability of the surface to connect the topography of the compositions and their biological affinity in vitro. The antibacterial properties and cell viability were essential in vitro studies to highlight the ability of these films to be suggested for medical applications. Furthermore, the in vivo study seems to be very useful, but it is not in the plan of this study.

(4)     What are the differences between (a), (b) and (c) in Figure 8. Are they taken at different areas on a same sample or on different samples? Figure 8 lacks scale bars.

Response: The scale bar of Figure 8 has been provided. Furthermore, an additional description has been provided through the caption of the figure to show that these figures were taken at different concentrations of the added films with the cells to evaluate the effect of the material’s concentrations on the mortality of the cells in vitro.

(5)     The manuscript should further be improved. Is the unit of the x-axis in Figure 7 “mg/ml” or “μg/ml”?

Response: The manuscript has been revised thoroughly now. The x-axis is “μg/ml”, now it has been corrected.

Reviewer 2 Report

The article entitled “Wound Dressings Containing Polysaccharide-based chitosan loaded with Metal oxide Nanohybrid: Synthesis, Characterization and Their Biological Activity Evaluation” is interesting and may be published in this journal after revision, as follows:

The methodology lacks references to the methods used.

The detailing of the characterization techniques needs to be complemented.

The GO XRD needs to be presented, as there was a greater effect after its addition. Indicate to whom the signs shown in the XRD figure are attributed.

The discussion/attribution of the FTIR bands needs to be better detailed in the text, despite the presence of the Table.

The EDX is a semi-quantitative data, however the graph presented did not indicate the presence of V and Yb. Authors need to review these data.

Graphene oxide is less hydrophilic than chitosan, how could an increase in hydrophilicity have occurred after insertion of GO?

The figures need to have their resolutions improved.

The conclusion needs to be improved, some data presented in the results needs to be included.

Author Response

Reviewer 2

The article entitled “Wound Dressings Containing Polysaccharide-based chitosan loaded with Metal oxide Nanohybrid: Synthesis, Characterization and Their Biological Activity Evaluation” is interesting and may be published in this journal after revision, as follows:

The methodology lacks references to the methods used.

Response: Appropriate references have been now mentioned in the materials and methods section.

The detailing of the characterization techniques needs to be complemented.

Response: Additional details have been provided through the characterization section to clarify the steps of using techniques.

2.3. Characterizations

X-ray diffraction (XRD) of (an analytical Pertpro with Cukα1, the Netherlands). The scan has been done from 10 degrees to 70 degrees. The step size was about 0.5 s. The wavelength of the X-ray beam is 1.54 Å. Fourier transforms infrared (FTIR) spectra were used for all samples by (Perkin - Elmer (FTIR) in the range of 4000 – 400 cm-1. The samples were prepared for the test by pressing the powdered/crashed film with KBr with a ratio of 20:1. Then the samples were fixed on the holder to be ready for the test. The morphology of the surface was investigated using a scanning electron microscope (SEM) (QUANTA-FEG250, The Netherlands). The operating voltage was around 8 kV. The thermal decomposition temperature of chitosan nanocomposites was determined by thermal gravimetric analysis (TGA) (STD Q600) from 25 to 800 ◦C at a heating rate of 10 ◦C/min in nitrogen medium.

The GO XRD needs to be presented, as there was a greater effect after its addition. Indicate to whom the signs shown in the XRD figure are attributed.

Response: The XRD caption has been modified to contain the signs shown in the graph and their meaning, while the representative peaks of GO are difficult to be distinguished among the peaks of metal oxides due to the low concentration of GO and the relative weakness of crystallinity for GO compared to the other metal oxide. Therefore, the presence of GO might be indicated by the FTIR peaks.

“Figure 1: XRD of CS with different ratios of nanocomposites Yb2O3 and V2O5 (* represents the peaks of V2O5, while the ● indicates the peaks of Yb2O3).”

The discussion/attribution of the FTIR bands needs to be better detailed in the text, despite the presence of the Table.

Response: The discussion of FTIR has been revised to be clearer as follows:

3.2. FTIR

The FTIR spectra of CS films with nanoparticles Yb2O3 and V2O5 are shown in Fig. 2 in the range of 500-4000 cm-1. It is used to determine the entire structure of each film based on the vibration bonding systems. For pure CS, the bands at 658, 894, and 994 cm-1 might refer to N-H bonds, glucosidic C-O-C stretching vibration, and C –O-C symmetrical stretching as in Table 1 [22-24]. The recorded bands 1091, 1152, and 1334 cm-1 may represent the  C-O stretching vibration and –OH group of the primary alcoholic group [12, 22, 25]. In addition, the determined bands at 1407, 1538, and 1555 cm-1 exhibit the –OH group of the primary alcoholic group, The symmetric bending of -NH+3, and amide II (N–H deformation present in the amino group) [22, 23, 26]. Bands at 2878 and 3397 cm-1 might assign to CH stretching vibration and the overlaps between primary amine -NH and OH group [23, 26]. In Yb2O3@CS the bands at 472, 611, 1512, and 1649 cm-1 were ascribed to Yb-O bonds, stretching vibrational modes of C–O bond, and C-O stretching vibration respectively [27, 28]. In the range of 3000-4000 cm-1, the O–H stretching vibrations were pretended [27]. The bands of 472, 611 and 613 cm-1 seem to represent the vibrational bond of Yb-O [27]. On the other hand, the vanadium oxide bonding system has been found in a band of 558 cm-1 [29]. It could be mentioned that the bonds of 658, 1334 and 1538 cm-1 might prove the contribution of CS through the composition.

The EDX is a semi-quantitative data, however the graph presented did not indicate the presence of V and Yb. Authors need to review these data.

Response: The EDX data has been revised thoroughly. The V and Yb are found but they seem to be with very low contribution, which might inhibit their peaks to be clearer.

Graphene oxide is less hydrophilic than chitosan, how could an increase in hydrophilicity have occurred after the insertion of GO?

Response: Additional statement has been provided through the discussion of the contact angle to clarify the prospective effects of the additional GO through the films as follows:

“The insertion of GO nanosheets through the composition might promote the formation of hydrogen bonding between the connected oxyanions over the graphene sheets with the graphene oxides, which has been mentioned in the discussion of the FTIR study. The formation of chemical bonds between the two constituents might develop rough surfaces as it was shown in the SEM graphs. These topographical changes might be essential to promote the hydrophilic affinity of the obtained composites.”

The figures need to have their resolutions improved.

Response: Some graphs have been replaced with additional ones to be with higher resolution.

The conclusion needs to be improved, some data presented in the results needs to be included.

Response: The conclusion has been modified and additional numerical results were provided to enrich the results that have been mentioned through the conclusion as follows:

  1. Conclusion

As the result of this study, chitosan (CS)-metal oxide (Yb2O3 and V2O5) nanohybrid (MONH) composite films were fabricated as wound bandages using the film-casting method in vitro. The micrographs of CS with Yb2O3 by SEM represented a rough surface with different pores has a diameter between 3-5 µm while the addition of V2O5 reduced the roughness and the scaffold becomes smooth, and the pores started to appear again after the addition of GO with diameter in the range of 0.5-2 µm. In addition, the thermal stability of the scaffolds was approved by the TGA curve where the complete degradation of the sample occurred at (462.9-597.3) º C with a total mass loss of about 79%. Moreover, the contact angle of CS with Yb2O3 was measured at 72.5 º, while the addition of V2O5 led to a slight increase in the hydrophilic behavior and the angle reached 68.2 º and the significant increase of hydrophilicity occurred when the GO nano-sheet was added and the angle was determined at 46.5 º. The cell viability has been examined versus normal cells, and it was found that the IC50 was found at 1997.2 µg/ml. In addition, the metal oxides incorporation through the scaffolds led to cell growth reaching 111.3% at a concentration obtained of 7.9 µg/ml. The antibacterial test has been carried out against E. coli and S. aureus. It was shown that the inhibition zone increased from 11.5±0.5 to 19.5±0.5 mm against E. coli for pure CS and Yb2O3/V2O5@CS, respectively. Furthermore, the inhibition zone of S. aureus increased from 12.5±0.5 to 19.0±1.0 mm for CS and Yb2O3/V2O5@CS.

Reviewer 3 Report

This work studied wound dressing made from chitosan loaded with metal oxide. In my opinion, the quality of this manuscript is not enough for publication in a peer-reviewed journal.

There was a lack of important experiments such as antibacterial properties test, O2 penetration test, physical properties such as flexibility test, etc …

The writing quality and presentation need to be enhanced.

Author Response

Reviewer 3

This work studied wound dressing made from chitosan loaded with metal oxide. In my opinion, the quality of this manuscript is not enough for publication in a peer-reviewed journal.

Response: The quality of the manuscript has been improved now. In addition, biological tests such as antibacterial behavior have been added. Hoping this meets the requirements of the journal level.

There was a lack of important experiments such as antibacterial properties test, O2 penetration test, physical properties such as flexibility test, etc …

Response: The antibacterial test has been carried out for all compositions. It has been added now in section 3.8.

3.8. Antibacterial behavior                 

The ability of the scaffolds to inhibit the prospective invasion of bacteria is one of the most important properties that should be available to avoid delaying the healing procedure. In this regard, the antibacterial behavior against gram-positive and gram-negative bacterial cells has been carried out for all under-investigated compositions to evaluate their strength against the two types of bacteria. As shown in Fig. 9, the potency of degenerating bacterial cells increased from the pure CS with the additional nanoparticles. The inhibition zone increased from 11.5±0.5 to 19.5±0.5 mm against E. coli for pure CS and Yb2O3 /V2O5@CS, respectively. On the other hand, the strength of the nanocomposites against S. aureus seems to be slightly higher than that of E. coli. The mortality of S. aureus increased from an inhibition zone of 12.5±0.5 to 19.0±1.0 mm for CS and Yb2O3 /V2O5@CS. The additional GO tends to plunge the efficiency of antibacterial whereas the inhibition zones decrease to 14.5±0.5 and 13.5±0.5 mm against both E. coli and S. aureus, respectively. The integration of Yb2O3 and V2O5 nanoparticles through the scaffolds might be the main parameter inducing the antibacterial potency due to the launching of reactive oxygen species with the toxic fragments against the bacterial cells. Nevertheless, pure GO has its own antibacterial properties, its addition does not reflect an improvement in the antibacterial efficacy. This unexpected behavior might be associated with the high potential of other nanoparticles to degenerate the bacterial cells compared with the GO. In other words, the additional GO on the account of Yb2O3 and V2O5 was not able to compensate for their potency against the bacterial cells with the same concentration.

Figure 9: Antibacterial effect of the scaffolds films of CS doped with different nanocomposites such as Yb2O3, V2O5, and/or GO against E. coli and S. aureus after 24 h of exposure.

The writing quality and presentation need to be enhanced.

Response: the English language has been revised through the whole manuscript.

Round 2

Reviewer 2 Report

The article can be accept in the present form.

Author Response

Comments and Suggestions for Authors

The article can be accepted in its present form.

We would like to thank the reviewer for his great efforts and for giving useful criticism of the article.

Reviewer 3 Report

After the revision of round 1, the quality of the manuscript has improved a lot. Although there are still lacking several characterizations, the work has meaning in the biomedical material field. I have several comments as follows:

1.       The introduction part should be remodified. Lines 45-47 and 48-49 need to be combined into one paragraph. Lines 50-69 should be combined into one paragraph. The last part of the introduction should be more detailed.

2.       The scale bar in figure 9 should be put on the image.

3.       I recommend the manuscript should go through the English checking one more time.

Author Response

Comments and Suggestions for Authors

After the revision of round 1, the quality of the manuscript has improved a lot. Although there are still lacking several characterizations, the work has meaning in the biomedical material field. I have several comments as follows:

We would like to thank the reviewer for his great efforts and for giving useful criticism of the article.

  1. The introduction part should be remodified. Lines 45-47 and 48-49 need to be combined into one paragraph. Lines 50-69 should be combined into one paragraph. The last part of the introduction should be more detailed.

Response:  Lines 45-47 have been combined into one paragraph.

Lines 50-69 have been combined in one paragraph.

The aim of the work has been enriched with more details as follows:

“In this study, novel wound dressings based on CS film containing hybrid nanocomposites including Yb2O3, V2O5, and GO are targeted. CS is thought to play a key role as a good matrix biomaterial for drug encapsulation. On the other hand, V2O5 is drugged in the CS film to fit with the targeted application as it has low toxicity, high anti-bacterial activity, and high biocompatibility, while Yb2O3 and GO are used to support the biological response of the scaffolds and to improve the antibacterial properties. These combinations of nanocomposites have been drugged in the CS film aim to improve the physical and chemical properties, bioactivity, and cell viability of the designed scaffolds to be appropriate for medical applications. The main structural and morphological features of the obtained films have been studied, while the surface wettability and the cell viability against normal cells were investigated. The antibacterial properties of the designed scaffolds of CS containing different contributions of Yb2O3, V2O5, and GO were evaluated against gram-positive and gram-negative pathogens to select the most optimum compositions for medical applications.”

  1. The scale bar in figure 9 should be put on the image.

Response: The scale bar now has been inserted in all images.

  1. I recommend the manuscript should go through the English checking one more time.

Response: The English language of the whole manuscript has been revised again. Hoping to meet the requirements of your journal.
